# Multiple electrolyte derangements among perioperative women with obstructed labour in eastern Uganda: A cross-sectional study

**Ritah Nantale**[1,2☯]*, **David Mukunya**[2,3,4☯], **Kenneth Mugabe**[2,5], **Julius N. Wandabwa**[2,5], **John Stephen Obbo**[2,6], **Milton W. Musaba**[2,5]

**1** Busitema University Faculty of Health Sciences, Department of Nursing, Mbale, Uganda, **2** Busitema University Centre of Excellence for Maternal Reproductive and Child Health (BuCEMaRCH), Mbale, Uganda, **3** Busitema University Faculty of Health Sciences, Department of Community and Public Health, Mbale, Uganda, **4** Department of Research, Nikao Medical Center, Kampala, Uganda, **5** Busitema University Faculty of Health Sciences, Department of Obstetrics and Gynaecology, Mbale, Uganda, **6** Department of Internal Medicine, Mbale Regional Referral Hospital, Mbale, Uganda

☯ These authors contributed equally to this work.
* ritahclaire24@gmail.com

**Data Availability Statement:** The dataset used for this analysis have been uploaded as supplementary information.

## Abstract

There is a dearth of information on the patterns of electrolyte derangements among perioperative women with obstructed labour. We measured the levels and patterns of electrolyte derangements among women with obstructed labour in eastern Uganda. This was a secondary analysis of data for 389 patients with obstructed labour, diagnosed by either an obstetrician or medical officer on duty between July 2018 and June 2019. Five milliliters of venous blood was drawn from the antecubital fossa under an aseptic procedure for electrolytes and complete blood analyses. The primary outcome was the prevalence of electrolyte derangements, defined as values outside the normal ranges: Potassium 3.3–5.1 mmol/L, Sodium 130–148 mmol/L, Chloride 97–109 mmol/L, Magnesium 0.55–1.10 mmol/L, Calcium (Total) 2.05–2.42 mmol/L, and Bicarbonate 20–24 mmol/L. The most prevalent electrolyte derangement was hypobicarbonatemia [85.8% (334/389)], followed by hypocalcaemia [29.1% (113/389)], then hyponatremia [18% (70/389)]. Hyperchloraemia [4.1% (16/389)], hyperbicarbonatemia [3.1% (12/389)], hypercalcaemia [2.8% (11/389)] and hypermagnesemia [2.8% (11/389)] were seen in a minority of the study participants. A total of 209/389 (53.7%) of the participants had multiple electrolyte derangements. Women who used herbal medicines had 1.6 times the odds of having multiple electrolyte derangements as those who did not use herbal medicines [Adjusted Odds Ratio (AOR): 1.6; 95% Confidence Interval (CI): (1.0–2.5)]. Having multiple electrolyte derangements was associated with perinatal death although this estimate was not precise [AOR 2.1; 95% CI: (0.9–4.7)]. Women with obstructed labour in the perioperative period have multiple electrolyte derangements. Use of herbal medicines in labour was associated with having multiple electrolyte derangements. We recommend routine assessment of electrolytes prior to surgery in patients with obstructed labour.

**Funding:** The authors received no specific funding for this work.

**Competing interests:** The authors have declared that no competing interests exist.

## Introduction

Obstructed labour is associated with life-threatening electrolyte derangements because of the protracted nature of the labour process that is characterized by long periods of inadequate oral intake [1], and anaerobic metabolism of glucose [2].

The associated maternal and perinatal morbidity and mortality is dependent on the nature of the imbalance in a specific electrolyte. For instance, an imbalance in maternal potassium levels is a known cause of life-threatening cardiac arrhythmias and cardiac arrest [2]. While imbalances in sodium especially hyponatremia during labour could result in cerebral edema which manifests as irritability, headache, nausea, vomiting, convulsions and coma that can easily be confused with preeclampsia and eclampsia [3,4]. In the postoperative period, electrolyte imbalances are a known cause of paralytic ileus especially after an emergency cesarean section [5]. Ileus is associated with post-cesarean maternal and perinatal morbidity because it interferes with early ambulation and oral intake, and consequently the initiation of breastfeeding [6]. Since the placenta is permeable to most of these ions either freely or through ion exchange mechanisms, the fetus is not spared from the effects of maternal electrolyte imbalance, which mirror those of the mother [7]. For instance, maternal hyponatremia is associated with an increased incidence of respiratory distress, hyperbilirubinemia, and neonatal convulsions in the postnatal period [3].

Moen *et al*. reported a 26% prevalence of hyponatremia among 287 patients labouring normally, in a Swedish hospital [3]. Among patients with obstructed labour, the prevalence of electrolyte imbalance is poorly studied [8,9]. In obstructed labour, serum bicarbonate is depleted in an attempt to buffer the excess acid ($H^+$ ions) produced as a by-product of anaerobic metabolism of glucose to produce energy in form of ATP for the contracting myometrium [10–12]. While the accumulation of potassium ions in the extracellular space (serum hyperkalaemia) arises from the exchange of intra-cellular $K^+$ ions for $H^+$ ions across the cell membrane to restore homeostasis [12,13]. In addition, obstructed labour in our setting is often preceded by a prolonged episode of labouring, restricted oral intake, inadequate intravenous fluid replacement by the providers, and at times consumption of local herbs that result in hypoglycaemia, dehydration, and hypernatremia [8,14].

Most patients with obstructed labour require surgical intervention to relieve the obstruction. Electrolyte imbalances are associated with poor surgical outcomes [15]. Intravenous fluid replacement is an important component of preoperative preparation to correct the associated dehydration and electrolyte imbalance [16]. Preoperative electrolyte measurement is not done routinely because of limited access to laboratory services and yet emergency cesarean section is the most common major surgical procedure performed on a daily basis. There is a dearth of information on the peri-operative patterns of electrolyte derangements among women with obstructed labour. We aimed to describe levels and patterns of electrolyte derangements among women with obstructed labour at Mbale regional referral hospital in eastern Uganda.

## Materials and methods

### Study setting and design

This was a descriptive cross-sectional study conducted among pregnant women screened for inclusion into a clinical trial at Mbale regional referral hospital in eastern Uganda. The rationale and design of this trial have been described elsewhere [17]. From July 2018 to June 2019, all patients diagnosed with obstructed labour (incident cases) were consecutively screened for inclusion in an ongoing trial that has since been published elsewhere [18]. Midwives in the delivery suite informed the study team of all participants diagnosed with obstructed labour

[17]. Two research assistants (RA) were available throughout the day and night to recruit all eligible women. As per the Ugandan clinical guidelines, the standard preoperative care for all patients diagnosed with obstructed labour includes prophylactic broad-spectrum antibiotics, at least 1.5 L of intravenous fluids, bladder emptying, administration of oxygen, and lying in a left-lateral position [19]. All these interventions were done after obtaining blood samples for electrolyte and haematological assessment.

## Study population

These were patients diagnosed with obstructed labour by either an Obstetrician or Medical Officer on duty using a definition of the American Association of Obstetricians and Gynaecologists (ACOG). In the first stage of labour, she should have cervical dilatation >6 cm with ruptured membranes, adequate contractions lasting >4 hours with no change in cervical dilatation or delay in the second active stage of labour (nullipara >2 hours, multipara >1 hour) with adequate uterine contractions. In addition, any two of the obvious signs of severe obstruction such as caput formation, severe moulding, Bandl's ring, subconjunctival haemorrhages, or an oedematous vulva [20].

## Sample size and sampling

We used the formula for determining sample size for frequency (single proportion) in a population as described in Open Epi, Version 3, open-source calculator [21]. We made the following assumptions; a confidence level of 95%, the proportion of the (prevalence of electrolyte derangements) in the population to be 50% since we could not find a similar published study, a design effect (DEFF) of 1, and power of 95%. It was estimated that a minimum sample of 384 participants would be sufficient but included all 389 women with obstructed labour who had been recruited after 12 months of data collection. We consecutively recruited all eligible incident patients diagnosed with obstructed labour until the calculated sample was attained.

## Eligibility criteria

We included all eligible patients with obstructed labour carrying singleton, term pregnancies (≥37 gestation age) in cephalic presentation. Whereas those diagnosed with other obstetric emergencies such as (antepartum haemorrhage, Pre-eclampsia, and eclampsia (defined as elevated blood pressure of at least 140/90 mmHg, urine protein of at least 2+, any of the danger signs and fits), premature rupture of membranes and intrauterine foetal death were excluded from this study. We also excluded all those with medical comorbidities such as diabetes, sickle cell, renal and liver diseases. Details are included in the published study protocol for the parent trial [17].

## Study variables

The dependent variable was the prevalence of electrolyte derangements defined as values outside the normal ranges at the end of the third trimester of pregnancy, categorized as either hypo for below or hyper for above and measured using standard laboratory methods. We adopted pregnancy-specific ranges from Abbasi *et al* [22] and converted these to the international system of units (SI) using an online converter [23]. For Potassium 3.3–5.1 mmol/L, Sodium 130–148 mmol/L, Chloride 97–109 mmol/L, Magnesium 0.55–1.10 mmol/L, Calcium (Total) 2.05–2.42 mmol/L, and Bicarbonate 20–24 mmol/L. Multiple electrolyte derangements was defined as having two or more electrolyte derangements. We considered two or more

electrolyte derangements since more than half of our sample were in this category. Perinatal death was defined as a fresh still birth or an early neonatal death at 7 days postnatal.

Other variables of interest were duration of labour in hours as reported by the patient, age in years, and physical characteristics such as weight, respiratory rate, pulse rate, blood pressure, foetal heart rate, and height.

### Study procedures

All the staff in the labour ward at Mbale regional referral hospital were sensitized about the ongoing study. The staff participated in giving information to patients about the study at the time of admission. At least two research assistants were available day and night to recruit all eligible patients diagnosed with obstructed labour by the attending Obstetrician or Medical Officer on duty. Written informed consent was obtained from each of the eligible patients by a trained research assistant before administering the questionnaire and collecting blood samples. Enrollment into the study was done as soon as obstructed labour was diagnosed before childbirth.

### Data collection

All the interviews were done in the participant's local dialect by trained research assistants, using an electronic interviewer-administered questionnaire running on an open data kit (ODK) platform [19]. On a daily basis, the principal investigator (PI) would access and review the data from the aggregate server for completeness.

### Blood sample collection quality control

Qualified and well-trained midwives working in the labour suite drew five millilitres (ml) of blood from each participant using a five millilitres syringe. After removing the epidermal needle and the tops of the vacutainers, two millilitres were emptied into a purple top vacutainer for a full blood count, blood grouping and cross-matching as part of the standard preoperative preparation for all patients with obstructed labour in the hospital. The remaining three millilitres were emptied into a red top vacutainer for blood chemistries including electrolytes, these are not part of the routine preoperative preparation for patients before caesarean section. Each blood sample was transported to the laboratory in an ice-cooled container within 30 minutes of collection. We monitored strict adherence to the standard operating procedures for collecting and handling participant samples through ongoing training and spot checks. We used MBN clinical laboratories that are accredited and certified both locally and internationally. They are involved in routine (monthly) internal and external quality control checks.

### Measurement of electrolytes

Serum electrolytes were measured using COBAS INTEGRA 400, which uses Ion Selective Electrodes (ISE) module. The ISE is an electro-mechanical device used to determine ionic concentrations in undiluted serum using electrons selective to different ions.

### Data management

The electronic questionnaire was coded with checks and skips for internal consistency. The data was uploaded to a password-protected aggregate server to which only the Principal Investigator had access. After the study, a soft copy of all the laboratory tests was obtained from MBN clinical laboratories as an excel spreadsheet. The Principal Investigator was assisted by a statistician to download data from the aggregate server into excel spreadsheets and merge the

two data sets. The data was then exported to Stata version 14.0 for further cleaning and analysis.

### Data analysis

We summarised baseline continuous variables using either means and standard deviations (SD) or medians and interquartile range (IQR). Whereas frequencies and percentages were used for the categorical variables. We used simple proportions to compute the prevalence of each electrolyte imbalance as a percentage with the corresponding 95% confidence interval. Logistic regression to determine factors associated with multiple electrolyte derangements among women with obstructed labour was done. Factors with a p-value less than 0.05 at multi-variable analysis were considered to be statistically significant.

### Ethics approval and informed consent

This study was approved by the Makerere University School of Medicine Research and Ethics Committee (#REC REF 2017–103) and the Uganda National Council for Science and Technology (HS217ES) approved the protocol. Administrative clearance was obtained from the Mbale regional referral hospital research and Ethics Committee (MRRH-REC IN-COM 00/2018). A written informed consent was obtained from each of the participants before enrolment. Pregnant adolescents below the legal age for consent of 18 years in Uganda were treated as emancipated minors who could give valid informed consent as per the national guidelines 25. A copy of laboratory results was availed in the patients' case file to aid the attending physicians in clinical decision-making regarding patient care.

## Results

We screened 451 women and enrolled 420 women diagnosed with obstructed labour. Fifteen declined consent, seven had a ruptured uterus and nine had concurrent medical comorbidities and other obstetric emergencies.

### Baseline characteristics of the study participants

Participants had a mean (standard deviation) age of 24.0 (6.2) years, majority 81.5% (317) were married. The patients generally had an elevated median capillary blood lactate level of 6.2 (3.3–13.5) mmol/L, a normal blood haemoglobin level 12.4 (1.9) and a high anion gap of 19 (13.4–23.5) mmol/L. They had a low median (interquartile range (IQR)) blood bicarbonate level of 13.2 (8.8–17.4) with a median duration of labour of 28 hours (18–43). About two-thirds (249/389) of respondents were admitted as referrals with obstructed labour and more than half (220/389) had used herbal medications during this labour. Almost all (90%) participants had a normal respiratory rate (16 to 24 breaths per minute). The details are in Table 1.

### Levels and patterns of electrolyte derangements among women with obstructed labour in eastern Uganda

Less than 5% (18/389) of the participants didn't have any electrolyte derangement, 47.7% (162/389) had one electrolyte derangement and 53.7% (209/389) had two or more electrolyte derangements (multiple electrolyte derangements). The most prevalent electrolyte derangement was hypobicarbonatemia [85.8% (334/389)] followed by hypocalcaemia [29.1% (113/389)]. Hyperbicarbonatemia [3.1% (12/389)], hypercalcaemia [2.8% (11/389)], and hypermagnesemia [2.8% (11/389)] were seen in a minority of the study participants.

**Table 1. Characteristics of the study participants.**

| Variable | N = 389 |
|---|---|
|  | **Mean (SD) / Frequency (%)** |
| **Maternal age, years** | 24.0 (6.2) |
| ≤19 | 110 (28.3) |
| 20–35 | 253 (65.0) |
| >35 | 26 (6.7) |
| **Marital status** | |
| Single | 72 (18.5) |
| Married | 317 (81.5) |
| **Maternal level of education** | |
| None | 9 (2.3) |
| Primary | 185 (47.6) |
| Secondary | 165 (42.4) |
| Tertiary | 30 (7.7) |
| **Religion** | |
| Christian | 263 (67.6) |
| Muslim | 122 (31.4) |
| Others | 4 (1.0) |
| **Maternal occupation** | |
| Salaried employee | 41 (10.5) |
| Business | 36 (9.3) |
| Subsistence farmer | 61 (15.7) |
| House wife | 195 (50.1) |
| Other | 56 (14.4) |
| **Place of residence** | |
| Urban | 41 (10.5) |
| Rural | 348 (89.5) |
| **Alcohol drinking** | |
| Yes | 12 (3.1) |
| No | 377 (89.9) |
| **HIV status** | |
| Positive | 6 (1.5) |
| Negative | 375 (96.4) |
| Don't know | 8 (2.1) |
| **Parity** | |
| Primigravida | 210 (54.0) |
| 2 to 4 | 125 (32.1) |
| 5+ | 54 (13.9) |
| **Labour duration (in hours)** | 28.2 (18.2–42.8) |
| <12 | 34 (8.7) |
| 12 to 18 | 63 (16.2) |
| >18 | 292 (75.1) |
| **Maternal height, cm** | 159.2 (8.3) |
| **Maternal weight, kg** | 64.8 (12.0) |
| **Maternal pulse rate, bpm** | 85 (18.7) |
| **Maternal systolic blood pressure, mmHg** | 122.4 (14.0) |
| **Maternal diastolic blood pressure, mmHg** | 75.3 (11.93) |
| **Maternal respiratory rate, bpm [median, IQR]** | 19.0 (18.0–22.0) |
| **Maternal haemoglobin level, mg/dl** | 12.4 (1.89) |

(*Continued*)

**Table 1.** (Continued)

| Variable | N = 389 |
|---|---|
| | **Mean (SD) / Frequency (%)** |
| **Maternal lactate, mmol/L** | 6.2 (3.3–13.5) |
| **Gestation age, completed weeks** | 38 (38–40) |
| **Mean fetal heart rate, bpm** | 137.9 (14.0) |
| **Anion gap*** | 19 (13.4–23.5) |
| **Ketones on urine dipstick** | |
| Negative | 214 (55.0) |
| One plus | 7 (1.8) |
| Two plus | 53 (13.6) |
| Three plus | 71 (18.3) |
| More than three plus | 44 (11.3) |
| **Referred** | |
| Yes | 249 (64.0) |
| No | 140 (36.0) |
| **Used herbal medicines** | |
| Yes | 220 (57.6) |
| No | 169 (43.4) |

Abbreviations; bpm- beats per minute.

Electrolyte derangements in the hypo (low) category were more pronounced than those in the hyper (high) category. The details are in Table 2.

Electrolyte derangements among women with obstructed labour, with the breakdown of (Fig 1) electrolytes by category (normal, high, low) for subgroups and (Fig 2) co-occurrence of electrolyte derangements. Fig 1 shows a ternary plot with balance of normal, high, and low levels for each of six serum electrolytes (bicarbonate, chloride, potassium, magnesium, sodium, and calcium), The six categories show the percent breakdown for each electrolyte group and the clustering of points in the upper left of the triangle indicates that most electrolytes were normal. Low electrolyte levels were observed most commonly, and elevated electrolyte levels were least common. Fig 2 shows electrolyte derangements and their co-occurrence. Horizontal bars (set size) show the prevalence of specific individual electrolyte derangement, while vertical bars (intersection size) show the co-occurrence. Overall, hypobicarbonatemia was the most common electrolyte derangement, followed by hypocalcemia, hyponatremia and hypochloraemia. hypercalcaemia, hypermagnesemia and hypernatremia were the least common electrolyte derangements overall. The most common presentation of derangements was hypobicarbonatemia alone (first vertical bar) [138 (35.5%)], followed by hypocalcaemia and hypobicarbonatemia (second vertical bar) [42 (10.8%)] then hypobicarbonatemia and hyponatremia (third vertical bar) [20 (5.1%)].

**Obstetric outcomes of women with obstructed labour in eastern Uganda.** A total of 41 out of 383 (10.7%) babies died (these were fresh still births or early neonatal deaths), 14/383 (3.7%) were sick and admitted to the neonatal unit and 328/383 (85.6%) didn't require any further treatment or attention. Having multiple electrolyte derangements [Adjusted Odds Ratio (AOR) 2.1; 95% Confidence Interval (CI): (0.9–4.7); P-value: 0.072] was associated with perinatal death although this estimate was not precise (S5 Table).

Amongst the mothers, 1/371 (0.3%) died, 8/379 (2.1%) suffered a morbidity and 370/379 (97.6%) didn't require any further treatment or attention.

**Table 2. Levels and patterns of electrolyte derangements among women with obstructed labour in eastern Uganda.**

| Electrolyte | Imbalance | Prevalence (N = 389) | | mmol/L | |
|---|---|---|---|---|---|
| | | n (%) | 95% CI | Median (IQR) | Q1—Q3 |
| **Sodium** | | | | 133.9 (4.9) | 131.2–136.1 |
| | Hyponatremia | 70 (18.0) | 14.5–22.5 | 127.3 (5.8) | 123.2–129 |
| | Normal | 319 (82.0) | 77.8–85.5 | 132.6 (4.3) | 132.6–136.9 |
| | Hypernatremia | 0 (0.0) | 0 (0.0) | 0 (0.0) | 0 (0.0) |
| **Potassium** | | | | 3.8 (0.7) | 3.8–4.21 |
| | Hypokalaemia | 47 (12.1) | 9.6–16.3 | 3.2 (0.2) | 3–3.23 |
| | Normal | 313 (80.4) | 75.7–83.6 | 3.9 (0.6) | 3.6–4.2 |
| | Hyperkalaemia | 29 (7.5) | 5.2–10.5 | 5.6 (0.4) | 5.4–5.8 |
| **Calcium** | | | | 2.14 (0.2) | 2.03–2.23 |
| | Hypocalcaemia | 113 (29.1) | 26.8–35.8 | 1.98 (0.1) | 1.89–2.02 |
| | Normal | 255 (65.6) | 60.7–70.1 | 2.2 (0.1) | 2.13–2.25 |
| | Hypercalcemia | 11 (2.8) | 2.3–6.3 | 2.45 (0.1) | 2.42–2.53 |
| **Magnesium** | | | | 0.71 (0.1) | 0.64–0.78 |
| | Hypomagnesemia | 23 (5.9) | 4.0–8.8 | 0.44 (0.2) | 0.34–0.52 |
| | Normal | 355 (91.3) | 88–93.7 | 0.71 (0.1) | 0.65–0.78 |
| | Hypermagnesemia | 11 (2.8) | 1.6–5.0 | 1.25 (0.5) | 1.13–1.62 |
| **Chloride** | | | | 101.8 (5.3) | 98.8–104.1 |
| | Hypochloraemia | 53 (13.6) | 10.5–17.4 | 95.3 (3.4) | 92.6–96.1 |
| | Normal | 320 (82.3) | 82–89 | 102.2 (4.3) | 99.9–104 |
| | Hyperchloremia | 16 (4.1) | 2.5–6.6 | 111.1 (2.4) | 109.9–112.3 |
| **Bicarbonate** | | | | 13.2 (8.6) | 8.8–17.4 |
| | Hypobicarbonatemia | 334 (85.9) | 82–89 | 12.1 (7.4) | 8.4–15.8 |
| | Normal | 12 (3.1) | 1.8–5.4 | 21.3 (1.7) | 20.4–22.1 |
| | Hyperbicarbonatemia | 43 (11.1) | 8.3–14.6 | 26 (1.7) | 25.1–26.3 |

Abbreviations: Q1- lower quartile; Q3- upper quartile; CI- confidence interval; IQR- Interquartile range.

**Factors associated with multiple electrolyte derangements among women with obstructed labor in eastern Uganda.** Herbal medicine use in labour was associated with having multiple electrolyte derangements. Women who used herbal medicines had 1.6 times the odds of having multiple electrolyte derangements compared to those who didn't use herbal medicines [AOR: 1.6; 95% CI: (1.0–2.5); P-value: 0.034]. Details are in Table 3.

## Discussion

A secondary analysis was done using a cross sectional study design to determine the levels and patterns of electrolyte derangements among women with obstructed labour in the perioperative period at Mbale regional referral hospital. More participants had hypo (low) as opposed to hyper (high) electrolyte derangements, with over 50% of the participants having multiple electrolyte derangements. This finding is in line with what was reported by Chebet *et al* among women with obstructed labour in Mulago National Referral hospital, more women had hypo (low) as opposed to hyper (high) electrolyte derangements [9].

Hypobicarbonatemia was the most prevalent electrolyte disorder, more than three quarters (¾) of the participants had a low median blood bicarbonate level of 12 mmol/L. This finding is not surprising because during labour, anaerobic breakdown of glucose to produce ATP leads to the accumulation of lactic acid which disintegrates to produce excess hydrogen ions that are

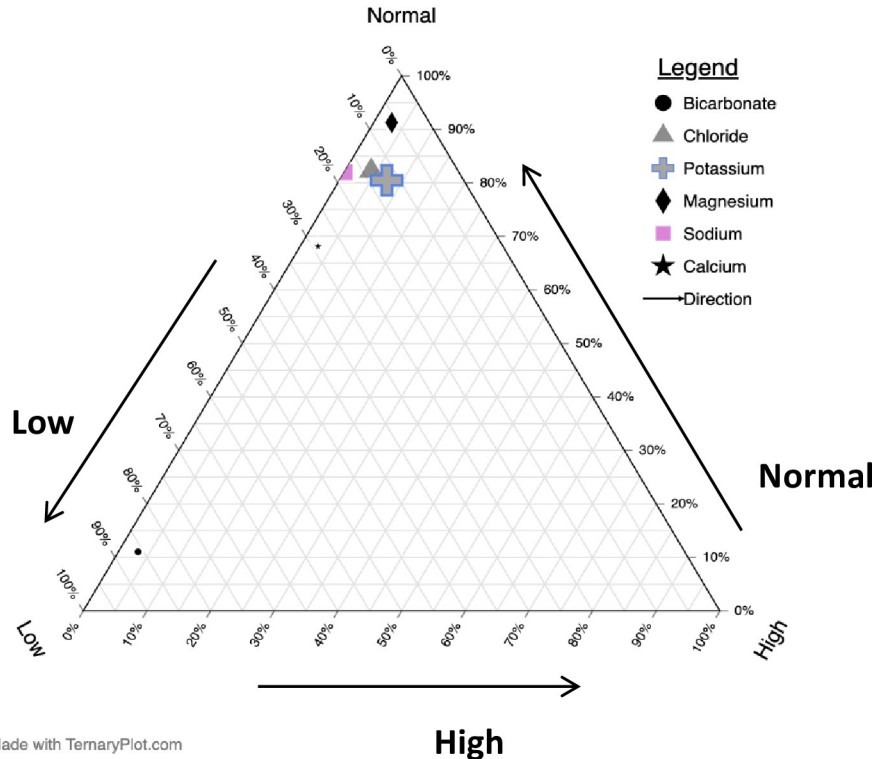

**Fig 1. Ternary plot with balance of normal, high, and low levels for each of six serum electrolytes (bicarbonate, chloride, potassium, magnesium, sodium, and calcium).**

mopped up by the bicarbonate buffer because it is one of the two important body buffer systems [24]. Although, our study had no comparison group of women who laboured without obstruction, the findings are similar with those reported from Nigeria. Ekanem *et al* [8] compared electrolyte imbalances among women with normal labour and those with prolonged labour and reported a mean bicarbonate level of 23.8 mmol/L among women with normal labour compared to 17.6 mmol/L among women with prolonged labour, all of which are higher than the 12 mmol/L finding in the current study. This difference could be partly explained by the fact that two thirds of the participants in our study were admitted as referrals with obstructed labour, which represents a more extreme category of patients with prolonged neglected obstructed labour. This category of patients is known to be more acidotic and consequently have a much lower blood bicarbonate levels [18]. This information is of clinical importance because prolonged uncorrected metabolic acidosis is known to cause low foetal pH, low Apgar scores and asphyxia in the newborn [25]. In the mother, acidosis impairs uterine contractility and this could increase the risk of primary postpartum haemorrhage due to uterine atony [10–12].

Hyponatremia was the only water balance disorder identified in this study. Lassey *et al.* reported hyponatremia in a case series of two parturient's secondary to prolonged labour and excessive oral hypotonic fluid consumption at home [26]. They suggested a likely mechanism of hyponatraemia to be endogenous oxytocin and excessive free water consumption in a setting of hypovolemia. In this study, we could not accurately ascertain the amount of fluid consumed during labour, the duration of labour and whether or not the patients had been vomiting because we recruited them after a diagnosis of obstructed labour had been made.

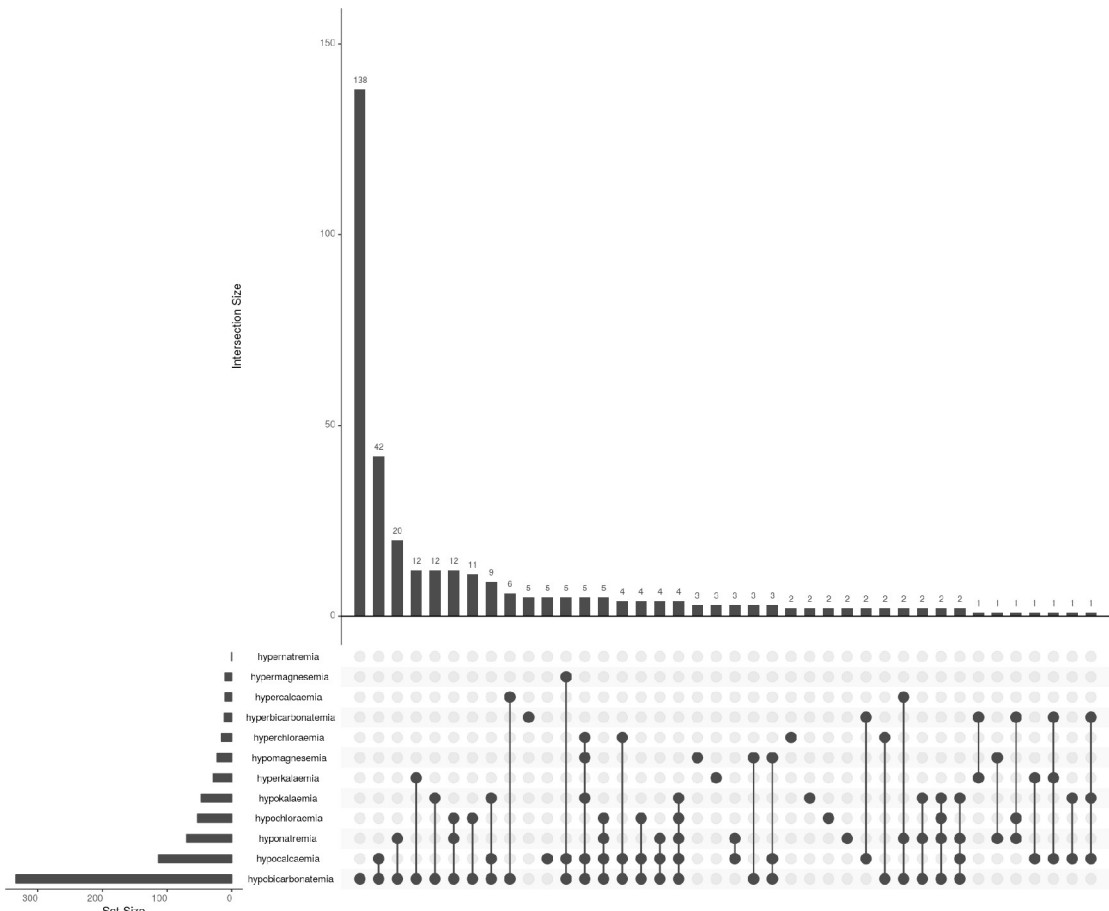

**Fig 2. Electrolyte derangements and their co-occurrence.**

Generally, the study participants were normotensive with a mean systolic and diastolic blood pressure of 122 and 75 mmHg respectively.

On the contrary, we expected hypernatremia to be the most common electrolyte disorder in this setting since obstruction is preceded by a period of prolonged labouring characterised by restricted intake of fluids and solids both orally and intravenously. These practices predispose labouring women to dehydration and electrolyte imbalances especially hypernatremia as well as hypoglycaemia [1]. Much as our finding of hyponatremia is consistent with findings from elsewhere [3,26], it cannot be fully explained by dilution alone due to excessive fluid intake/replacement during labour that is not a common practice in this setting. An alternative explanation might be the activation of specific V2 vasopressin receptors in the kidney. During labour, the stimuli for vasopressin release such as hypoosmolality, pain, fear and stress are abundant and they may be pronounced in obstructed labour [3]. It is possible that these stimuli combine to mimic the syndrome of inappropriate ADH secretion because of the similarity in structure between oxytocin and vasopressin [3]. Although labour its self is not a known cause of inappropriate ADH secretion. A study by Chebet et al., 2022 in Uganda also found out that hyponatraemia is prevalent among women with obstructed labour [9]. Hyponatraemia needs to be corrected because it is associated with several maternal and neonatal risks that may be life threatening [26].

**Table 3. Factors associated with multiple electrolyte derangements among women with obstructed labor in eastern Uganda.**

| Variable | Multiple electrolyte derangements N (%) | COR (95% CI) | P-value | AOR (95% CI) | P-value |
|---|---|---|---|---|---|
| **Maternal age** | | | | | |
| ≤19 | 55 (26.3) | 1 | | | |
| 20 to 35 | 139 (66.5) | 1.2 (0.8–1.9) | 0.386 | 1.5 (0.9–2.4) | 0.154 |
| >35 | 15 (7.2) | 1.4 (0.6–3.2) | 0.481 | 1.3 (0.4–3.7) | 0.649 |
| **Parity** | | | | | |
| Primigravida | 116 (55.5) | 1 | | | |
| 2 to 4 | 60 (28.7) | 0.7 (0.5–1.2) | 0.200 | 0.7 (0.4–1.1) | 0.110 |
| 5+ | 33 (15.8) | 1.3 (0.7–2.3) | 0.438 | 1.2 (0.5–2.5) | 0.705 |
| **Marital status** | | | | | |
| Single | 38 (18.2) | 1 | | | |
| Married | 171 (81.8) | 1.0 (0.6–1.7) | 0.858 | 0.9 (0.5–1.5) | 0.678 |
| **Religion** | | | | | |
| Christian | 135 (64.6) | 1 | | | |
| Muslim | 72 (34.4) | 1.4 (0.9–2.1) | 0.160 | 1.4 (0.9–2.2) | 0.178 |
| Others | 2 (1.0) | 0.9 (0.1–6.8) | 0.958 | 1.1 (0.1–8.2) | 0.961 |
| **Alcohol drinking** | | | | | |
| Yes | 4 (1.9) | 1 | | | |
| No | 205 (98.1) | 2.4 (0.7–8.1) | 0.162 | 2.3 (0.6–8.1) | 0.200 |
| **HIV status** | | | | | |
| Positive | 2 (1.0) | 0.5 (0.1–4.5) | 0.535 | 0.6 (0.1–6.2) | 0.670 |
| Negative | 203 (97.1) | 1.2 (0.3–4.8) | 0.817 | 1.4 (0.3–5.9) | 0.655 |
| Don't know | 4 (1.9) | 1 | | | |
| **Referred from a lower health facility** | | | | | |
| No | 77 (36.8) | 1 | | | |
| Yes | 132 (63.2) | 0.9 (0.6–1.4) | 0.706 | 0.8 (0.5–1.3) | 0.432 |
| **Herbal medicines use in labour** | | | | | |
| Yes | 127 (60.8) | 1.4 (0.9–2.2) | 0.071 | **1.6 (1.0–2.5)** | **0.034** |
| No | 82 (39.2) | 1 | | | |
| **Labour duration** | | | | | |
| <12 | 16 (7.7) | 1 | | | |
| 12 to 18 | 37 (17.7) | 1.6 (0.7–3.7) | 0.272 | 1.8 (0.7–4.3) | 0.191 |
| >18 | 156 (74.6) | 1.3 (0.6–2.6) | 0.482 | 1.3 (0.6–2.7) | 0.521 |

COR: Crude Odds Ratio; AOR: Adjusted Odds Ratio; Bold: Statistically significant at a p-value<0.05.

About one third of our participants had hypocalcaemia (total calcium). We did not find any published papers from a similar setting for direct comparisons. However, this prevalence is high and it may not solely be attributable to nutritional deficiencies and none supplementation during pregnancy [27]. Ionised calcium plays a central role in the excitation, contraction and coupling mechanisms in the myometrium [12,14]. Since obstructed labour is characterised by strong and regular contractions, probably the demand for ionised calcium is high and this could result in hypocalcaemia. Our results have to be interpreted cautiously because we measured the total and not the ionised calcium that is required for uterine muscle contraction. Hypocalcaemia in labour could be attributed to labouring polypnea [28]. However, in our study almost all (90%) participants had a normal respiratory rate. The association between

hypocalcaemia and obstructed labour needs further study by comparing with controls that experience normal labour.

Hypokalaemia was the most prevalent disorder of serum potassium, this was lower than the 20% prevalence reported among hospitalised patients, despite the fact that we adopted a wider normal range for the serum potassium levels (3.3 to 5.1 mmol/L) [24]. In line with our findings, a study by Chebet et al., 2022 also reported hypokalaemia to have been the most prevalent disorder of serum potassium in women with obstructed labour [9]. This is probably as a result of an intracellular shift in exchange for the excessive hydrogen ions generated from the lactic acid produced in obstructed labour. In this study, the prevalence of hyperkalaemia was slightly higher than what was reported in the study by Ekanem et al in Nigeria [8]. Derangements in potassium, especially hyperkalaemia is associated with life threatening arrythmia. Perioperatively, this is important information for clinicians to know because these patients will need anaesthesia before surgical intervention to relieve the obstruction.

Herbal medicine use in labour was associated with having multiple electrolyte derangements. Similarly, a study by Chebet *et al*., 2022 found an association between electrolyte imbalances and herbal medicine use in this same population [9]. Herbal medicines are of unknown safety, concentrations and efficacy levels and may cause kidney injury resulting from the toxicity imparted by the untested drugs thus leading to electrolyte imbalances [29]. Chebet et *al*., 2022 also reveals that herbal medicine use involves excessive intake of water, often with limited intake of salt and solutes which potentially limits the capacity of the kidney to excrete water leading to an electrolyte imbalance [9].

Water and electrolyte balance are affected by a number of factors that were not assessed in this study because the participants were only recruited after a diagnosis of obstructed labour had been confirmed. We could not accurately ascertain the amount, volume and type fluid used since the onset of labour. The inability to measure these factors; amount, volume and type fluid used since the onset of labour could have introduced a non-differential misclassified bias in our study. Additionally, majority of them were admitted as referrals with obstructed labour without proper documentation. Our assessment of the duration of labour and consumption of local herbs is based on patient reports which may not be accurate. Furthermore, we didn't establish the exact types of herbal medicines used by the participants.

Methodological considerations;

This study was adequately powered to study the various electrolyte derangements and it is probably among the first in this population of patients from the same setting. However, this study had some limitations. The lack of a comparison group of women who laboured successfully without obstruction makes it difficult to attribute these electrolyte derangements to obstructed labour. However, a previous study from Nigeria reported differences in electrolyte derangements among those with and without obstructed labour. Nonetheless, the results of this study can be generalised to other regional referral hospitals within Uganda and other similar settings in sub-Saharan Africa.

## Conclusion

Women with obstructed labour in the perioperative period have multiple electrolyte derangements, especially if they have used herbal medicines in labour. The derangements were more pronounced in the hypo (low) compared to the hyper (high) category. Clinicians need to know that life threatening electrolyte derangements are common among women with obstructed labour in the perioperative period.

## Supporting information

**S1 Table. Factors associated with hyperkalaemia and hypokalaemia among women with obstructed labor in eastern Uganda.**
(DOCX)

**S2 Table. Factors associated with hyponatremia among women with obstructed labor in eastern Uganda.**
(DOCX)

**S3 Table. Factors associated with hypobicarbonatemia among women with obstructed labor in eastern Uganda.**
(DOCX)

**S4 Table. Factors associated with hypocalcaemia among women with obstructed labor in eastern Uganda.**
(DOCX)

**S5 Table. Factors associated with perinatal death among women with obstructed labor in eastern Uganda.**
(DOCX)

**S1 Data. Stata dataset that includes all variables analysed for this manuscript.**
(DTA)

## Acknowledgments

We thank the study participants for accepting to be part of the study and the research midwives for working tirelessly to accomplish this task on time namely; Ms. Auma Prosscovia, Ms. Nandutu Sarah Waterah, Mrs. Atim Ketty Ojwar, Ms. Alibo Elizabeth, Ms. Sarah Talyewoya and Ms. Jessica Muduwa.

## Author Contributions

**Conceptualization:** Ritah Nantale, David Mukunya, Kenneth Mugabe, Julius N. Wandabwa, John Stephen Obbo, Milton W. Musaba.

**Data curation:** Ritah Nantale, David Mukunya, Kenneth Mugabe, Julius N. Wandabwa, John Stephen Obbo, Milton W. Musaba.

**Formal analysis:** Ritah Nantale, David Mukunya, Milton W. Musaba.

**Funding acquisition:** Milton W. Musaba.

**Investigation:** Ritah Nantale, David Mukunya, Kenneth Mugabe, Julius N. Wandabwa, Milton W. Musaba.

**Methodology:** Ritah Nantale, David Mukunya, Kenneth Mugabe, Julius N. Wandabwa, John Stephen Obbo, Milton W. Musaba.

**Project administration:** Ritah Nantale, David Mukunya, Kenneth Mugabe, Julius N. Wandabwa, John Stephen Obbo, Milton W. Musaba.

**Resources:** Ritah Nantale, David Mukunya, Kenneth Mugabe, Julius N. Wandabwa, John Stephen Obbo, Milton W. Musaba.

**Software:** Ritah Nantale, David Mukunya, Kenneth Mugabe, Julius N. Wandabwa, John Stephen Obbo, Milton W. Musaba.

**Supervision:** Ritah Nantale, David Mukunya, Kenneth Mugabe, Julius N. Wandabwa, John Stephen Obbo, Milton W. Musaba.

**Validation:** Ritah Nantale, David Mukunya, Kenneth Mugabe, Julius N. Wandabwa, John Stephen Obbo, Milton W. Musaba.

**Visualization:** Ritah Nantale, David Mukunya, Kenneth Mugabe, Julius N. Wandabwa, Milton W. Musaba.

**Writing – original draft:** Ritah Nantale, David Mukunya, Kenneth Mugabe, Julius N. Wandabwa, John Stephen Obbo, Milton W. Musaba.

**Writing – review & editing:** Ritah Nantale, David Mukunya, Kenneth Mugabe, Julius N. Wandabwa, John Stephen Obbo, Milton W. Musaba.

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
