## [Decision Letter · Decision Letter 0]

8 Feb 2023

PGPH-D-22-01944

Multiple electrolyte derangements among perioperative women with obstructed labour in eastern Uganda: A cross-sectional study

Dear Dr. Nantale,

Thank you for submitting your manuscript to PLOS Global Public Health. After careful consideration, we feel that it has merit but does not fully meet PLOS Global Public Health’s publication criteria as it currently stands. Therefore, we invite you to submit a revised version of the manuscript that addresses the points raised during the review process.

I would like to sincerely apologise for the delay you have incurred with your submission. It has been exceptionally difficult to secure reviewers to evaluate your study. We have now received three completed reviews; the comments are available below. The reviewers have raised significant scientific concerns about the study that need to be addressed in a revision. Please revise the manuscript to address all the reviewer's comments in a point-by-point response in order to ensure it is meeting the journal's publication criteria. Please note that the revised manuscript will need to undergo further review, we thus cannot at this point anticipate the outcome of the evaluation process.

We look forward to receiving your revised manuscript.

Kind regards,

Miquel Vall-llosera Camps

Staff Editor

Journal Requirements:

State the initials, alongside each funding source, of each author to receive each grant.

2. Please provide separate figure files in .tif or .eps format.

4. In the online submission form, you indicated that "Data supporting the results reported in this manuscript can be obtained from the corresponding author on request.". All PLOS journals now require all data underlying the findings described in their manuscript to be freely available to other researchers, either 1. In a public repository, 2. Within the manuscript itself, or 3. Uploaded as supplementary information.

Reviewers' comments:

Reviewer's Responses to Questions

**Comments to the Author**

1. Does this manuscript meet PLOS Global Public Health’s publication criteria? Is the manuscript technically sound, and do the data support the conclusions? The manuscript must describe methodologically and ethically rigorous research with conclusions that are appropriately drawn based on the data presented.

Reviewer #1: Yes

Reviewer #2: Yes

Reviewer #3: Partly

2. Has the statistical analysis been performed appropriately and rigorously?

Reviewer #1: Yes

Reviewer #2: Yes

Reviewer #3: I don't know

3. Have the authors made all data underlying the findings in their manuscript fully available (please refer to the Data Availability Statement at the start of the manuscript PDF file)?

Reviewer #1: Yes

Reviewer #2: No

Reviewer #3: Yes

4. Is the manuscript presented in an intelligible fashion and written in standard English?

Reviewer #1: Yes

Reviewer #2: Yes

Reviewer #3: Yes

5. Review Comments to the Author

Reviewer #1: Good paper to emphasize the life threatening electrolyte derangement which are common among women with obstructed labour in the perioperative period. However, the study had some limitations by means of the lack of a comparison group of women who laboured successfully without obstruction makes it difficult to attribute these electrolyte derangements to obstructed labour.

Reviewer #2: 1. The study reported by the authors add to the literature what is already known about the effect of protracted labour on electrolyte levels. Since the baseline data as reported by the authors are already known in the literature, one would have expected that the authors would have considered extending the what is known about the subject by including at least birth outcomes to try to link the electrolyte derangements to maternal and fetal outcomes. However, this was not done even though the authors had ample time to have collected such data. If such data is available, authors should include this in the revised manuscript to bring out some novelty to the data presentation. The authors write on page 17 “. This information is of clinical importance because prolonged uncorrected metabolic acidosis is known to cause low fetal pH, low, Apgar scores and asphyxia in the newborn.” This is suggestive that maternal and/or fetal outcomes are dependent on these electrolyte levels. Thus, why the authors did not include such data collection is not so clear.

2. Since the most prevalent electrolyte disorder was hypobicarbonatemia, as reported by the authors on page 16, “Hypobicarbonatemia was the most prevalent electrolyte disorder, more than three quarters (¾) of the participants had a low median blood bicarbonate level of 12 mmol/L”, the authors should at least consider regression analyses for factors associated with this electrolyte derangement since it is most pronounced.

3. Also, since about one third of our participants had a hypocalcaemia (total calcium), author should consider predisposition to this as well.

4. Also, could the authors consider including as supplementary data analyses of factors predictive of participants having 3, 4, 5 or 6 electrolyte derangements? This would be an important addition since there might be reason(s) why one person may develop combinations of these electrolyte imbalances.

Reviewer #3: This is a cross-sectional study analyzing metabolic derangement among women undergoing urgent cesarean section for obstructive labour.

You state that hypocalcemia (total calcium) is common in your population. Is this possibly related to a pre-eclamptic condition of some of the mothers in your cohort? Also, labouring women (and even more so women with obstructed labour) present polypnea. Can this be related to observed calcium derangements? Please comment.

Factors listed in line 321-326 represent strong limitations to this study, and must be discussed in the limitation section.

A thorough English mothertongue revision should also be performed before resubmission.

6. PLOS authors have the option to publish the peer review history of their article (what does this mean?). If published, this will include your full peer review and any attached files.

**Do you want your identity to be public for this peer review?** For information about this choice, including consent withdrawal, please see our Privacy Policy.

Reviewer #1: **Yes: **Dr Pradeep Kumar Dabla

Reviewer #2: **Yes: **Patrick Adu

Reviewer #3: No

---

## [Decision Letter · Decision Letter 1]

13 Apr 2023

PGPH-D-22-01944R1

Multiple electrolyte derangements among perioperative women with obstructed labour in eastern Uganda: a cross-sectional study

Dear Dr. Nantale,

Thank you for submitting your manuscript to PLOS Global Public Health. After careful consideration, we feel that it has merit but does not fully meet PLOS Global Public Health’s publication criteria as it currently stands. Therefore, we invite you to submit a revised version of the manuscript that addresses the points raised during the review process.

Two reviewers reassessed the manuscript. Reviewer 3 would like additional clarification on some of their queries. Please ensure that you provide a point-by-point response to their comments.

We look forward to receiving your revised manuscript.

Kind regards,

Hanna Landenmark

Staff Editor

Journal Requirements:

Additional Editor Comments (if provided):

Reviewers' comments:

Reviewer's Responses to Questions

**Comments to the Author**

1. If the authors have adequately addressed your comments raised in a previous round of review and you feel that this manuscript is now acceptable for publication, you may indicate that here to bypass the “Comments to the Author” section, enter your conflict of interest statement in the “Confidential to Editor” section, and submit your "Accept" recommendation.

Reviewer #1: All comments have been addressed

Reviewer #3: (No Response)

2. Does this manuscript meet PLOS Global Public Health’s publication criteria? Is the manuscript technically sound, and do the data support the conclusions? The manuscript must describe methodologically and ethically rigorous research with conclusions that are appropriately drawn based on the data presented.

Reviewer #1: Yes

Reviewer #3: Yes

3. Has the statistical analysis been performed appropriately and rigorously?

Reviewer #1: Yes

Reviewer #3: I don't know

4. Have the authors made all data underlying the findings in their manuscript fully available (please refer to the Data Availability Statement at the start of the manuscript PDF file)?

Reviewer #1: Yes

Reviewer #3: Yes

5. Is the manuscript presented in an intelligible fashion and written in standard English?

Reviewer #1: Yes

Reviewer #3: Yes

6. Review Comments to the Author

Reviewer #1: Ok to publish

Reviewer #3: I can't find in the revised manuscript a discussion addressing the points I raised in my previous review. Is hypocalcemia

possibly due to a pre-eclamptic condition? If yes, in what percentage of the set of patients presented by authors? Also, is hypocalcemia related to labouring polypnea of women? Please comment.

7. PLOS authors have the option to publish the peer review history of their article (what does this mean?). If published, this will include your full peer review and any attached files.

**Do you want your identity to be public for this peer review?** For information about this choice, including consent withdrawal, please see our Privacy Policy.

Reviewer #1: **Yes: **The authors have made an study efforts to observe the women with obstructed labour in the perioperative period and shown having multiple electrolyte derangement, especially if they have used herbal medicines in labour which were more pronounced in the hypo (low) category. These observations will help in clinical evaluations further.

Reviewer #3: No

---

## [Editor Report · Decision Letter 2]

11 May 2023

Multiple electrolyte derangements among perioperative women with obstructed labour in eastern Uganda: a cross-sectional study

PGPH-D-22-01944R2

Dear Ms. Nantale,

We are pleased to inform you that your manuscript 'Multiple electrolyte derangements among perioperative women with obstructed labour in eastern Uganda: a cross-sectional study' has been provisionally accepted for publication in PLOS Global Public Health.

Best regards,

Julia Robinson

Executive Editor